



Hydrology and
Earth System
Sciences

# Rainbow color map distorts and misleads research in hydrology – guidance for better visualizations and science communication

**Michael Stoelzle[1] and Lina Stein[2]**

[1]Faculty of Environment and Natural Resources, University of Freiburg, Freiburg, Germany
[2]Department of Civil Engineering, University of Bristol, Bristol, UK

**Correspondence:** Michael Stoelzle (michael.stoelzle@hydro.uni-freiburg.de)

**Abstract.** Nowadays color in scientific visualizations is standard and extensively used to group, highlight or delineate different parts of data in visualizations. The rainbow color map (also known as jet color map) is famous for its appealing use of the full visual spectrum with impressive changes in chroma and luminance. Besides attracting attention, science has for decades criticized the rainbow color map for its nonlinear and erratic change of hue and luminance along the data variation. The missed uniformity causes a misrepresentation of data values and flaws in science communication. The rainbow color map is scientifically incorrect and hardly decodable for a considerable number of people due to color vision deficiency (CVD) or other vision impairments. Here we aim to raise awareness of how widely used the rainbow color map still is in hydrology. To this end, we perform a paper survey scanning for color issues in around 1000 scientific publications in three different journals including papers published between 2005 and 2020. In this survey, depending on the journal, 16 %–24 % of the publications have a rainbow color map and around the same ratio of papers (18 %–29 %) uses red–green elements often in a way that color is the only possibility to decode the visualized groups of data. Given these shares, there is a 99.6 % chance to pick at least one visual problematic publication in 10 randomly chosen papers from our survey. To overcome the use of the rainbow color maps in science, we propose some tools and techniques focusing on improvement of typical visualization types in hydrological science. We give guidance on how to avoid, improve and trust color in a proper and scientific way. Finally, we outline an approach how the rainbow color map flaws should be communicated across different status groups in science.

## 1 Why does the rainbow color map distort and mislead scientific visualizations?

Colorful visualizations are deeply integrated in science communication. In hydrology, visualization of water fluxes like precipitation, evapotranspiration, discharge or percolation and terms like green and blue water, humidity and aridity, or flood and drought are subjects of the daily hydrologists work. Our presentation of patterns, relationships, compositions, distributions and comparisons of multivariate datasets is often multifaceted. And they are most often encoded with color (Wong, 2011a). This is first of all reasonable as human perception is dominated by visual perception (70 % compared to 30 % by the other senses). The human eye can recognize around 10 million unique colors but only 30 shades of grey (Kreit et al., 2013). Today computer software and freely available programming tools like R or Python simplify the use of color and color gradients in color maps. The rise of online-only journals reduced the necessity for a good perception of black–white printed graphs or papers. Although colorful graphs and maps can be created with a few clicks, the development of a compelling visualization is a complex task.

In terms of correct encoding, visual mappings such as position, length, angle, direction, area and volume rank higher in efficiency and accuracy than color (e.g., Wong, 2010). In other words, the human eye is stronger in encoding data that is mapped in a bar plot or scatterplot than in a colorful heat map. The encoding accuracy of color maps has primarily been criticized when it comes to the rainbow color map. This color map uses all wavelengths of the visible spectrum between 380 and 750 nm, impresses with high lightness and

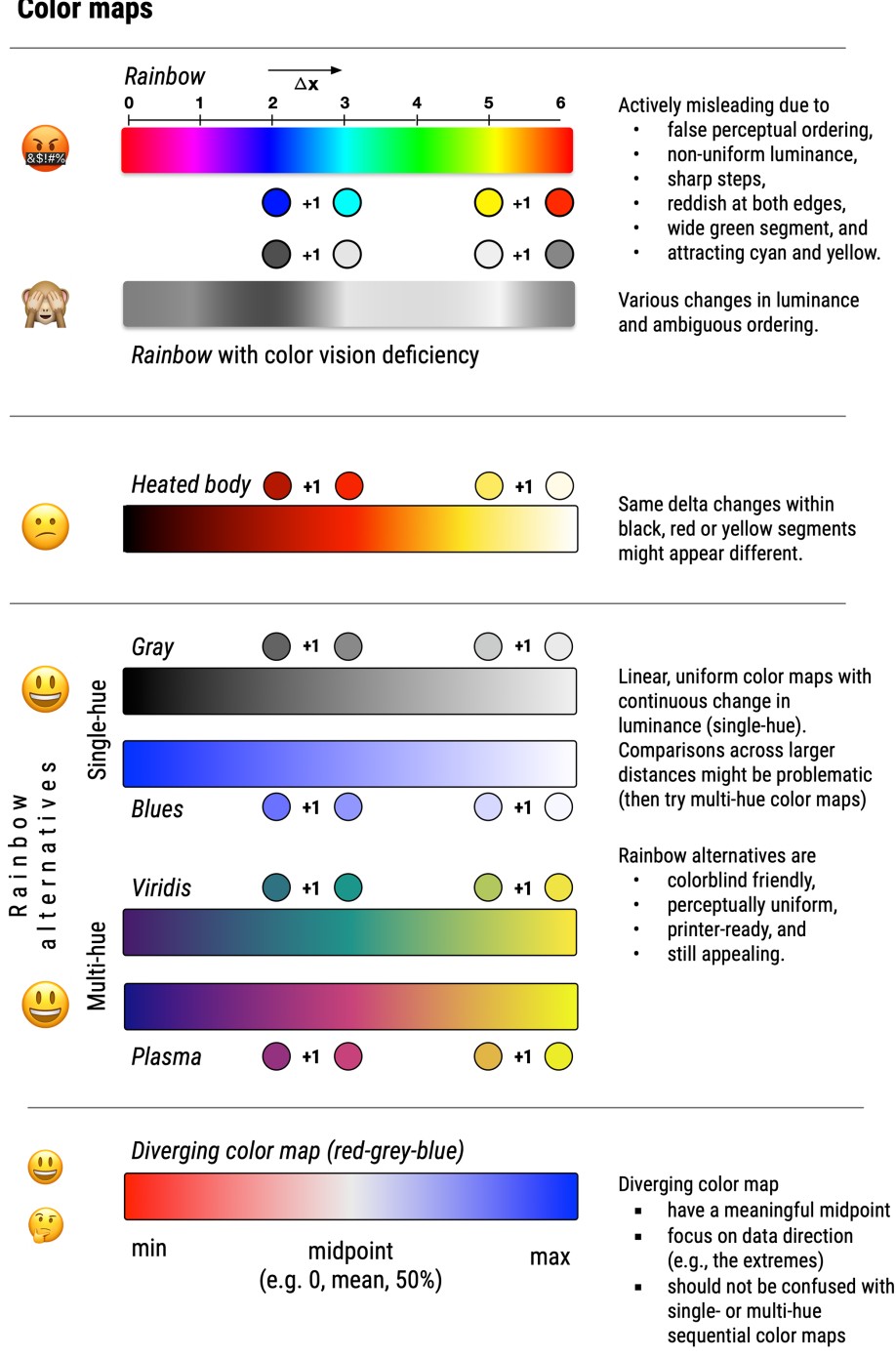

**Figure 1.** Comparison of different color maps along an arbitrary scale (0–6). The same delta changes in $x$ values (denoted with +1) are not represented uniformly in the rainbow or heated body color map due to unordered luminance. Alternatives are monochromatic, single-hue color maps (greys, blues) or perceptually uniform designed multi-hue color maps like viridis or plasma. Diverging color maps highlight data extremes and the data direction (e.g., positive/negative data values) if a meaningful midpoint is apparent in the data. Visualization inspired by the literature (Crameri et al., 2020; Wong, 2011b).

chroma, encapsulates the most saturated colors, and hence looks at first sight very appealing and eye-catching (Fig. 1). In the past, the rainbow (or jet) color map or red and green colors for sequential data were often the software standard, having a wide use in the scientific communities and hence also in publications. For example, in former versions of the statistics software R (version 3.x) the pre-set color map used black, red, green and dark blue as the first four colors and failed numerous colorblind checks.

In general, there are two main reasons why the rainbow color map in scientific visualization is "(still) considered harmful" (Borland and Taylor, 2007). First, color vision deficiency (CVD) affects the perception of 8 %–10 % of the male and 0.4 %–0.5 % of the female population, depending on earth regions, and thus up to 4 % of the world's population (Geissbuehler and Lasser, 2013; Nuñez et al., 2018; Pramanik et al., 2012). CVD shares are given for Caucasian people and might be lower among other ethnic groups. The simultaneous and side-by-side use of red and green as in the rainbow color map obstructs an unbiased access to the visualization for these people. Putting an 8 % CVD ratio into perspective, a Caucasian male team of one editor and two reviewers during a paper review has a chance up to 22.1 % that at least one person has a CVD (Wong, 2011b).

Secondly, the rainbow color map attracts attention but is weak in representing data in a scientifically correct way (Fig. 1). This affects all people, even those with normal color vision. The same Euclidean distances in mapping or the same data ranges in continuous or binned variables are not equally represented by a rainbow color map (Crameri et al., 2020; Sharma and Trussell, 1997). Especially for data comparisons over a wider distance in the color map, the distorted colors of the rainbow impede reliable judgments (Liu and Heer, 2018). Abrupt changes of lightness and saturation often lead to an unintended focus on some sections of the data range (Thyng et al., 2016; Wong, 2011a). The high lightness of the yellow, cyan or magenta segments in the rainbow color map makes it difficult to perceive a consistent color and data value ordering (Kovesi, 2015). Also, high and low values could be confused if both are represented by reddish colors at the edges of the rainbow color map. The color map distorts the data representation if the change in value is not visually commensurate with the change in color (Wong, 2010, 2011c). Discordant false coloring may lead to visual errors up to 7.5 % of the total displayed data variation (Crameri et al., 2020). For example, research has also shown that replacing a rainbow color map with a perceptually uniform color map could identify hidden structures in mapping (Rogowitz et al., 1996). In perceptually uniform color maps, the delta change in color is equal to delta change in data. Comparisons of rainbow color maps and perceptually uniform color maps in cartographic mapping have demonstrated that rainbow colors can emphasize strong gradients where actually smooth data variation is apparent (Fig. 3 in Thyng et al., 2016). Empirical judgment of different quantitative color maps has hence identified the rainbow color map as perceptually much slower and more error-prone compared to single-hue color maps or perceptually uniform designed multi-hue color maps (Liu and Heer, 2018).

A thoughtful and scientifically correct color map should allow for all types of dichromatic views (i.e., color vision deficiency) and unambiguous perception of the displayed data. As it is now scientific standard and best practice to avoid any rainbow or rainbow-derivate color map (Crameri et al., 2020), we want to challenge the use of rainbow color maps in hydrological science by analyzing the status quo of rainbow visualizations in hydrological and environmental publications. In this study we evaluate the use of rainbow and red–green color use in the journal *Hydrology and Earth System Sciences* (*HESS*) in preprints and publications in 2020 and their use over time (2005–2020). We then compare the results to two other journals which cover different disciplines. Finally, we discuss alternatives for using color overall in scientific publications and how to improve and trust the use of color.

## 2 Meta-analysis with paper survey

### 2.1 How often is the rainbow color map used in scientific visualizations?

There are discrepancies between theoretically known scientific standards and the de facto use of the rainbow color map. A non-representative survey of presentations and posters at EGU 2018 (European Geosciences Union) found 60 % included at least one rainbow scale figure (McNeall, 2018). Compared to publications, visualizations are even more essential for poster presentations and conference talks as less time and text is available to present the research results. The appealing effect of rainbow color maps is often used as eye-catcher along the poster walls. Due to the peer-review process, we hypothesized that the ratio of rainbow color maps in publications should be notably lower than 60 %. If there is a considerable number of scientific publications with rainbow color maps, is there at least a decreasing tendency towards fewer rainbow color maps in recent years?

First, we examined all preprints that were published in October 2020 ($n = 36$) in the journal *Hydrology and Earth System Sciences* (*HESS*). We found 25 % of these preprints having at least one graph or map with a rainbow color map. Three of these rainbow preprints colored 70 %–80 % of all figures with a rainbow color map. Interestingly, the median author number of the rainbow preprints was five, suggesting that rainbow-colored visualizations are not necessarily seen as a critical issue during manuscript preparation and internal submission processes. We then consulted the author guidelines from the journal (https://www.hydrology-and-earth-system-sciences. net/submission.html, last access: 29 July 2021) to check what

kind of color recommendations are given for the authors. We found in total two occurrences of the term "color" on the web page. In the section "Figures and tables", recommendations for high-quality graphics are given with "For maps and charts, please keep color blindness in mind and avoid the parallel usage of green and red. For a list of color scales that are illegible to a significant number of readers, please visit ColorBrewer 2.0".

In a second step after the preprint analysis, we evaluated if the review process reduces the use of rainbow-colored visualizations. We did that by screening in total 263 peer-reviewed papers published in *HESS* in the year 2020. To our knowledge, no systematic review of rainbow color maps in environmental journals exists so far. The journal guidelines of *HESS* also recommend to avoid green and red colors side by side in visualizations. We therefore classify the papers into four groups also considering pure black–white papers:

   A. black–white paper without use of any color,

   B. paper that has no rainbow-colored visualization or supports distinction with additional elements,

   C. paper that has at least one visualization with rainbow-related coloring or use of green and red elements without a good chance to separate these elements,

   D. paper that has at least one rainbow-colored visualization (graph or map).

That means a graph with a red and green boxplot could be classified as acceptable (class B) as often axis labels explain the boxplot groups. A graph with two lines (red and green; see Fig. 6 "Original") encoding continuous variables over time without any annotations more than the legend is classified as rainbow-related (class C). If a paper has a rainbow color map visualization, then potential misuse of red/green is not further counted in our statistics.

TS1 The majority of 168 (64 %) papers in 2020 have not included any rainbow-colored visualization. For 58 papers (22 %), we found at least one graph or map that uses explicitly the rainbow color map. In 37 papers (14 %) we classified at least one graph as "rainbow-related" (e.g., use of the spectral color map) or identified red–green data encodings without a good chance to distinguish different lines or points. Summarizing these color issues, our survey shows that around 36 % of the publications in *HESS* in 2020 had visualizations that are not scientifically correct, not perceptually uniform, and unaccessible or hard to access for around 4 % of the readership due to color vision deficiency. This indicates that the awareness of misleading color choice is rather low during the publication process for both authors and reviewers. This was further confirmed by evaluating reviewer comments of articles published in 2020. We searched for keywords "blind", "color", "colour", "green" and "deficiency". Of the 263 articles published, nine reviewer comments (3.4 %) mention necessary improvements regarding

color or problems with readability of the graph. Only two comments specifically address the issue a red–green color scale will have for some readers. In reaction, one published article changed the color scale to orange–blue instead of red–green. The other article changed the color map in most graphs to an improved color map but also did change one color map to rainbow color map. For a further three articles, the reviewer criticized readability of the graph. In reaction, one article changed a red–green coloring to red–black. The other two did not change their figures and continued the use of red–green or rainbow colors. This demonstrates that reviewers and editors are not sufficiently aware of this problem, which is also justified by comparable rainbow color map shares in preprints and peer-reviewed, published papers.

We then extended the survey for all papers published in *HESS* in 2015, 2010 and 2005 to better understand if there is or was a tendency towards more or fewer rainbow-colored visualizations in scientific publications (Fig. 2, Table A1). Nearly half of the examined 800 papers (from 2005, 2010, 2015 and 2020) have been classified with critical use of ambiguous or not colorblind-friendly color maps (Fig. 2). Survey results indicated that the ratio of papers with rainbow or rainbow-related color maps has been stable between 2010 and 2020 but markedly increased between 2005 and 2010. From 2005 to 2010 there was a clear increase in color use (from 56 % to 82 %), and black and white papers dropped by 26 %. The survey of 797 papers results in 9 % pure black and white papers; 47 % of chromatic papers showed no color issues, and 44 % of all papers have either used a rainbow color map in at least one visualization and/or have embedded a visualization with red–green issues. Two cross checks with 30 and 50 randomly chosen papers led to minor deviations in color classification due to the personal judgment of our reviewer team (three people). However, a high fraction of rainbow classified papers from the main survey were also classified as rainbow papers in the cross checks, with 86 % (6 out of 7 papers) and 92 % (12 out of 13 papers) agreement.

## 2.2   Is the use of rainbow color maps a journal- or discipline-specific artifact?

To answer this question, we screened in total 200 additional publications from different disciplines (e.g., environmental science, biology) in the renowned journals *Nature Scientific Reports* and *Nature Communications*. For *Nature Scientific Reports*, we looked at the top 100 of most downloaded papers in the section "Earth Science" in 2019 (accessible via https://www.nature.com/collections/agegihhehi, last access: 29 July 2021). On the corresponding website, graphical thumbnails are given to preview the research findings. Here we found 10 out of 100 thumbnails have rainbow color maps or rainbow-related coloring. Going into more detail we scanned the 100 papers and found that 26 % of the papers used rainbow color maps in at least one figure (class D) and 18 % have figures with potential red–green issues in color

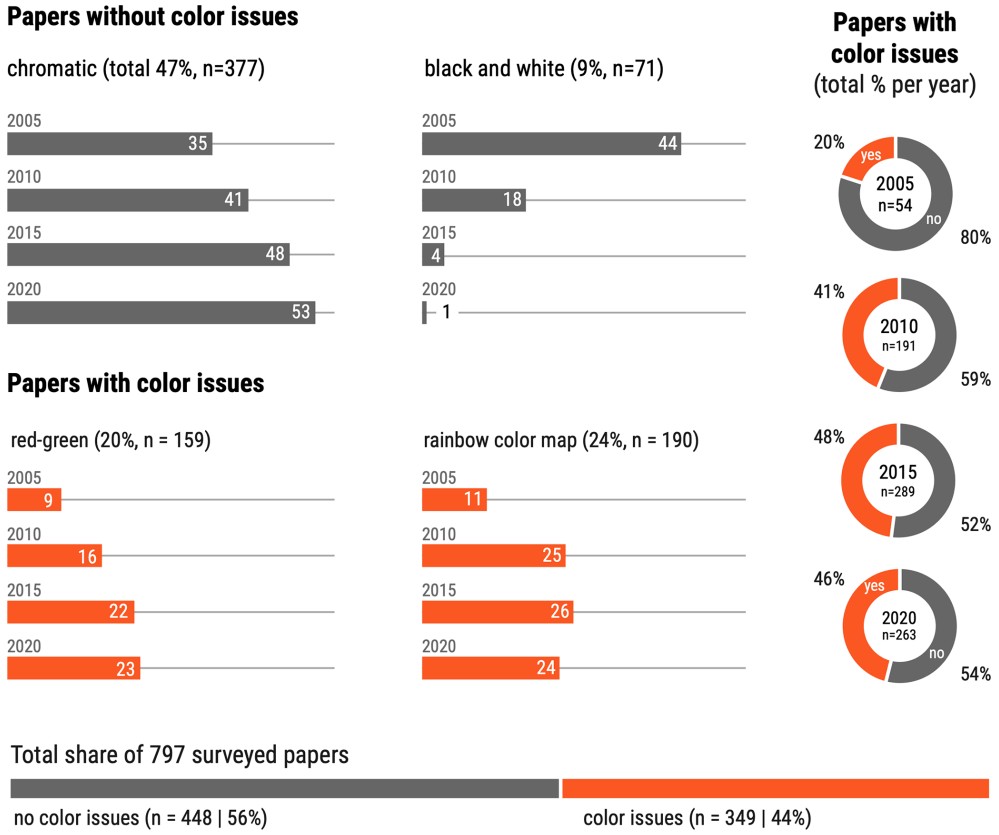

**Figure 2.** Proportion of surveyed papers without and with color issues. In total 797 peer-reviewed papers from *Hydrology and Earth System Sciences* (*HESS*) with different shares for the years 2005, 2010, 2015 and 2020 were analyzed. Missing percent up to 100 is due to rounding. In 2020 only papers that were published before 1 November are considered.

perception (class C). We also counted 6 % of the surveyed papers in this journal as black and white papers (class A). However, more than half of the investigated papers in *Nature Scientific Reports* have been classified to be color-issue free (Fig. 3). In *Nature Communications* (survey of 100 papers published between 4 and 6 November 2020), we found no papers in class A (black–white), 55 % of papers without color issues (class B), 29 % of papers that used red and green (class C), and 16 % with a rainbow color map (class D) in at least one visualization (Fig. 3).

Altogether, we surveyed 997 scientific papers from three journals (published between 2005 and 2020, with 46 % of papers from 2019 and 2020) and found 23.7 % of all papers have at least one visualization colored with a rainbow color map. The ratio of misused red–green color combinations is most likely even higher than reported here as red–green issues in rainbow papers are not separately counted in the statistics. However, our results revealed a considerably lower ratio of rainbow color maps compared to Borland and Taylor (2007). Their survey from the 2001 through to 2005 IEEE Visualization Conference proceedings found around 50 %–60 % of papers having at least one rainbow visualization. Putting our survey results into perspective, when picking randomly 5 (10) papers one still has a 75.0 % (93.7 %) chance to encounter at least one paper with a rainbow visualization. If red–green issues are also considered, then the chance of at least one visual problematic paper in a selection of 5 (10) papers is 94.4 % (99.6 %). Our survey suggests a co-occurrence between the decline of black and white papers and the emergence of papers with color issues. For 2005 we find that 20 % of the papers published in *HESS* have color issues, and for 2020 we find that 47 % of papers have color issues. In the same time the share of black and white papers dropped from 44 % to less than 1 % in 2020.

When analyzing the effect of color issue awareness among author teams, we clearly see that a higher number of authors did not necessarily lead to a lower share of color-issue-free papers (Fig. 4). Chromatic papers without color issues had a 40 %–50 % share regardless of how many authors were included in these publications. Black and white papers are mostly published as single-author papers or by teams of two authors (27 % and 14 %, respectively). However, only 4 % of all papers are published with visualizations in black and white, and the share has been decreasing since 2005 (Fig. 2). The share of papers with rainbow color issue for single-author papers is around 9 %. In contrast, consid-

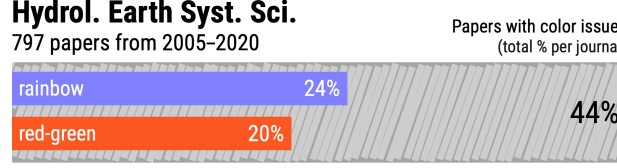

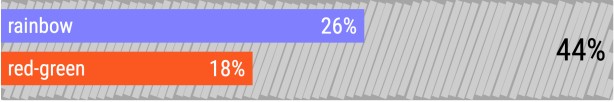

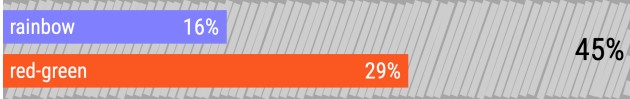

**Figure 3.** Share of papers with at least one visualization with red–green issues or a rainbow color map across different journals.

ering multiple-author publications, around 20 %–30 % of the papers have rainbow color issues.

We speculate that single-author publications are more often composed by senior authors preferring simpler but often clearer visualizations where color is not (primarily) needed to encode the data. Another feasible explanation is that in former years papers were written by fewer authors than today. Average author number per paper in our *HESS* paper survey was 3.72 in 2005 and 5.06 in 2020. Text mining analysis of paper title terms (see Appendix, Fig. A1) suggests that often studies with spatial analyses or cartographic maps have an above-average chance that a visualization with color issues is embedded. Here papers with title terms such as "terrestrial", "map(s)" or "mapping", "radar", "satellite", and "region" or "regional" have in between 73 % and 92 % of the cases (Fig. A1) a rainbow-colored figure or a figure with red–green issues in the paper.

## 3   Four steps to go beyond the rainbow color map

To overcome the need for the rainbow color map, we present four suggestions to avoid, improve, trust and communicate color in scientific visualizations. The central questions for these four steps are given in Fig. 5 as techniques to improve the use of color and to communicate misuse of color. Kelleher and Braswell (2021) published a comprehensive introduction on how visualization challenges of environmental data can be assessed in a broader sense.

## 3.1   Avoid color – learning from black and white visualizations

Taking inspiration from older papers with black and white visualization is a valuable approach to identify potential improvement of colored visualizations. In former times technology and/or computer software did not allow for the same use of color in visualizations as today. Years ago, colored pages in visualizations were also additionally charged by the publishers. Today color is often the first choice for data encoding in visualizations (Wong, 2011a), but colors are also often used without any reason. Despite the fact that the human eye can differentiate millions of colors, Stauffer et al. (2015) stated that only a small number of different hues can be processed for important classification tasks (search and distinguishing). Healey (1996) showed that only around seven different hues can be found accurately and rapidly on a map or cartographic application. On maps, and also heat maps, the neighboring colors and the distance between two colored elements bias the perception of data variability (Brychtová and Çöltekin, 2017). If so, visualizations with extensive color use should be revised to reduce the number of colors or redesigned using other graphical encodings. Structure, hierarchy, clarity and completeness can instead be used to create an appealing look in a figure instead of color.

Some examples in the surveyed and other literature illustrate potential ways of doing this: visualization of model biases with a grey color gradient (Schaefli et al., 2005); black and white map shadings (Milly, 1994); response time distributions of different catchments with lines with various greys, thicknesses, and line types (Roa-García and Weiler, 2010); monthly regime curves of different climate models by lines with different line types and additional point symbols to highlight a specific baseline (Kingston and Taylor, 2010); monochromatic mappings and cumulative fluxes (Campbell et al., 2015); stacked bar charts with a sequential grey color scale (Sunyer et al., 2015); and various point shapes to represent groups (Burden et al., 2019; Sunyer et al., 2015) or visualizations with direct labeling instead of legends (Hoellein et al., 2019).

Hydrology as a science particularly uses line charts to illustrate change over time, e.g., in streamflow analysis. We thus suggest to find visual inspiration from black and white papers to demonstrate how color could be avoided or reduced. At the same time, specific aspects of a visualization can be explicitly highlighted for the reader (Fig. 6). If a single technique is not sufficient to improve the visual statement, then a combination of techniques could be also feasible, e.g., lines with various widths and types, additional overlaid points on the lines, or direct labels to highlight specific lines such as the baseline or the mean (Fig. 6). Especially direct labeling could improve the clarity of the (line) graphs leading to fewer cluttered graphs due to additional white space when the legend box is removed (Fig. 6e and f). Text elements, rich in contrast, give guidance for the reader

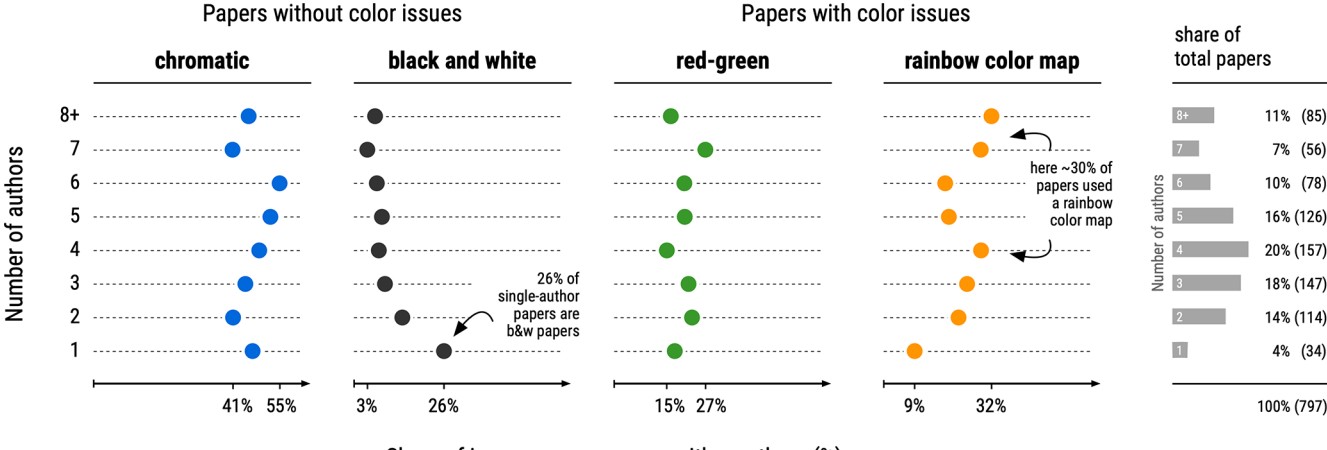

**Figure 4.** Composition of color issues with a focus on number of authors. Labels of *x* axes show minimum and maximum values among the categories. Each row adds up to 100 %.

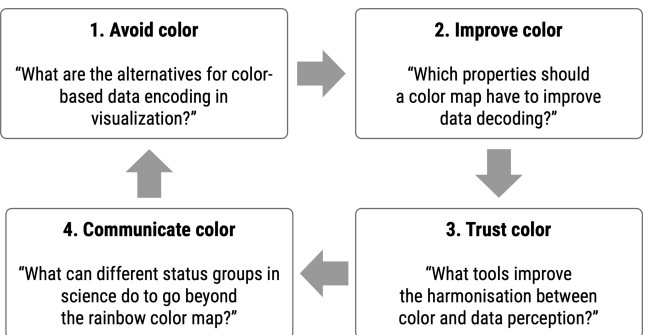

**Figure 5.** Circular flow of central questions to improve and communicate color in scientific visualizations. The different tasks are discussed in the Sect. 3.1 to 3.4. Table 1 provides a checklist for the different tasks (properties, tools, etc.).

to comprehend the story and support people with low vision to easily identify the major elements of the visualization.

Scientists have the possibility to use other visual encodings than color such as position (e.g., scatterplots), length (e.g., bar charts), or different point shapes or line widths (Fig. 6a–d) to increase the perception of data variation in the visualization (Kelleher and Wagener, 2011; Wong, 2010). Furthermore, variation in transparency (Fig. 6c) could also improve the clarity of visualization especially when data points are plotted above each other like in large sample hydrology or other scatterplots. If data encoding is shifted from color to other mappings such as position or line width, a targeted use of one or two colors is a strong technique to highlight specific parts of the visualization such as baselines, extremes, averages, or specific periods or regions (Fig. 6g). Improved visualization should be accompanied by elaborative figure captions (Fig. 6i), and all authors are asked to create self-explanatory figures that are fully understandable solely with the information in the caption (Rougier et al., 2014). Informative, story-telling graph titles may enhance data visualizations (Wanzer et al., 2021), although they are still unpopular in the scientific community.

Besides improvements in single visualizations, splitting figures and maps into different subplots (i.e., facets in Fig. 6h) allows for multiple views on the story of a visualization (Shoresh and Wong, 2012). Here the facets or sparklines (Streit and Gehlenborg, 2015) replace different colors in a single plot. A common technique is to present all data points in all facets as a background data variation and then use color or point shapes to highlight specific data groups in the single facets (Gnann et al., 2019, 2020). This multi-facetted view might be valuable for Budyko curve analyses, visualization of different model runs, catchment comparisons or to highlight different distributions of data groups along one axis (histograms, density plots, area charts), on two axes (scatterplots) or if grouping in stacked visualizations (bar or area charts) is encoded by color. For example, dense scatterplots with a lot of overplotting, like storage–discharge plots in hydrological recession analysis (Stoelzle et al., 2013) or in large sample hydrology, could profit from faceting as then data encoding is shifted from color to position.

Another important issue in hydrological science is heat maps as they allow us to visualize a third variable in a two-dimensional coordinate system. Heat maps notably rely on color encoding and thus need an appropriate color map and a meaningful order of the data (categories) on rows and columns. Rethinking the order of rows and columns in heat maps often reveals that an alphabetical or a chronological order is not the best choice. Similarity and clustering of categories may help out here (Gehlenborg and Wong, 2012a), and clustering can be done by splitting a single heat map into multiple heat maps. As color perception in a heat map depends on colors of neighboring cells, varying stroke color or

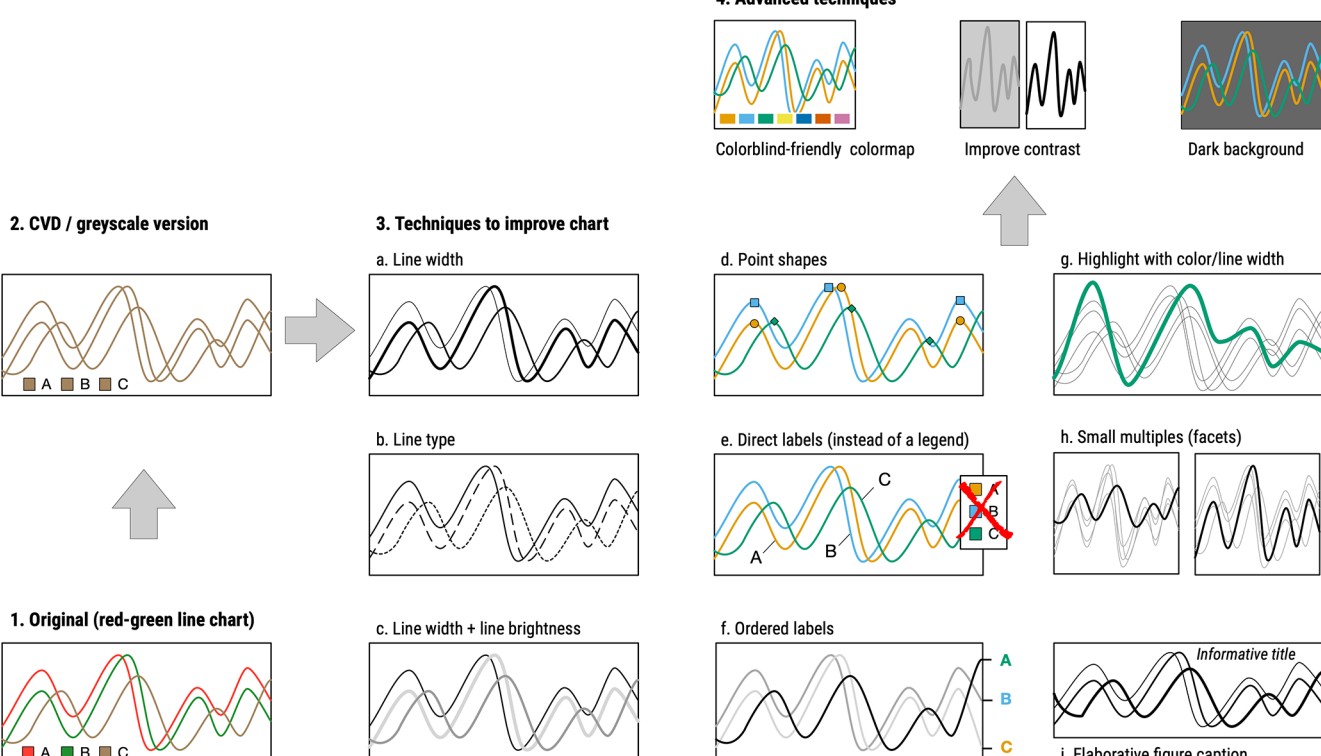

**Figure 6.** Recommendations to improve colorful line graphs or line graphs with red and green lines next to each other (steps 1 and 2). Improvements can be achieved (steps 3 and 4) by adjustment of line width, type or brightness **(a–c)**; adding additional points or labels to lines **(d–f)**; focusing on one specific line **(g)**; using small multiples to allow for easier comparison **(h)**; and ensuring that the figure is fully self-explanatory with a precise and complete figure caption and an informative graph title **(i)**. Additional, more advanced techniques (4.) are colorblind-friendly color maps, increased contrast and dark(er) backgrounds (e.g., increasing contrast to highlight lines during a presentation) to further improve (line) charts.

line width around the tiles will improve the data perception (Supplementary Fig. 3 in Crameri et al., 2020). Adding numeric values as text to heat map tiles reduces the ambiguity of color perception. Parallel coordinate plots might be a valu-
5 able alternative to heat maps by shifting the data encoding from color to position (Gehlenborg and Wong, 2012a). Multivariate data can be split into several two-dimensional visualizations to improve the clarity of the visualization (Gehlenborg and Wong, 2012c).

10 But there is another very important aspect in advancing data visualizations. Extensive use of color and perceptually ineffective color maps like the rainbow color map impede not only people with CVD but also affect people with low or reduced vision. The World Health Organization (WHO) states
15 that "at least 2.2 billion people have a vision impairment" (World Health Organization, 2019). Although a lot of those people have received professional eye care, a reduction of visual acuity could impede access to overloaded and cluttered visualizations with less contrast. It is important to recognize
20 that the group of people with low or limited vision or visibility is much larger (up to 28 % of world's population) than the group of people with CVD. To visualize for people with low

vision, high contrast and supportive text elements or pointers are most important. Visualization should hence be improved with high text and element contrasts, annotations, the 25 ease of horizontal labeling without line breaks, clear figure structure with elaborated hierarchy and focus within the visualization (Tufte, 1983). Gestalt principles such as similarity, closure, proximity and common regions help to achieve grouping and partitioning in graphs. The idea of data–ink ra- 30 tio (Tufte, 1983) could help to remove cluttered, non-data elements and allow for more white space as this helps to focus on important parts of the visualization. Fewer, thinner or removed grid lines also increase the data–ink ratio and sharpen the view on the data. 35

In a second step, changing perspective from authors to publishers, the community needs more advanced tools than static PDF or printed papers (Vandemeulebroecke et al., 2019). If articles are more and more published in HTML format, interactive visualizations could allow for multiple per- 40 spectives by data zoom, selection and layering (e.g., Gehlenborg and Wong, 2012c). Interactive visualizations give the possibility for people with low vision or CVD to select supportive elements like tooltips during mouse hovering or by

highlighting selected elements and give direct labeling and annotations. Using alt attributes in HTML can be used to specify alternative text if a visualization cannot be rendered or perceived. Interactive elements on basis of HTML and R packages are, for example, dygraphs (Vanderkam et al., 2018), with sliders to select specific time periods in time series analysis or leaflet maps (Cheng et al., 2021), giving the possibility to zoom into maps and to select different background layers (i.e., leading to higher contrast for people with vision impairment). Authors can also accompany their data analysis in a paper with an online available data dashboard such as Shiny apps (Chang et al., 2020) for further data exploration. Such efforts could be also beneficial for people with other limited (cognitive) capabilities (i.e., low visualization literacy or blind people) as there are possibilities of speech- or touch-based interaction with non-static visualizations (Lee et al., 2020).

## 3.2 Improve color – what are alternatives to the rainbow color map?

Although black and white visualizations could inspire a thoughtful revision of colored figures or maps, removing all color is not always the best choice (Kelleher and Braswell, 2021). Depending on data dimensions, the use of color is sometimes unavoidable. Typically, we want to use color to convey data and also to create a figure or map that looks appealing. Crameri et al. (2020) presented a thoughtful decision tree explaining how color depending on data types should be used in visualizations. They differentiated between the direction of color gradients to encode higher or lower values according to the chosen background color (light or dark). Dark(er) backgrounds have recently gained more and more attention in the visualization community as a possibility to increase the contrast of visualizations (Crameri et al., 2020). Although dark figure backgrounds make for an unusual sight in articles, the increase in contrast might be appropriate for presentations, helping people with vision impairments. If a single-hue color map such as the blue color map in Fig. 1 is not sufficient to encode data by color, then blended-hue color maps are a feasible solution. For example, to visualize depth below and elevation above sea level a combination of two monochromatic color maps with a reasonable midpoint or breakpoint can be used (Gehlenborg and Wong, 2012b). If small data variations in a continuous color map are not needed, then a discrete or binned color map with fewer but more distinguishable colors support faster decoding of color and data variation. Thoughtful breaks for color binning and better boundaries among important data ranges or regions give guidance for the reader. Here the proximity and orientation of the legend could also help to gain undistorted allocation of the presented data range. Depending on the degree of data variability on $x$ and $y$ axes, the legend could be horizontally or vertically aligned to the graph.

In summary, finding an alternative for the rainbow color map can be seen as a relatively straightforward process when two main aspects of color maps are considered. Firstly, there are many perceptually uniform color maps available that are well documented, professionally designed and do consider people with CVD. As a starting point, the online tool ColorBrewer 2.0 (https://colorbrewer2.org, last access: 29 July 2021) can be used to find appropriate, colorblind-friendly color maps. Such color maps are also often available in the visualization software or programming languages for example the viridis (Garnier, 2018) or scico (Pedersen and Crameri, 2020) packages in R or the seaborn library in Python (Waskom, 2021). The MATLAB software uses the "parula" color map as a default, but also other color maps can be set up. Regardless of the used software, a thoughtful color map should be perceptually uniform, colorblind friendly, strong in greyscale conversion and – if possible – pretty and appealing to attract the reader (Fig. 7). Famous examples are the Okabe color map (Okabe and Ito, 2008), the viridis color map (Garnier, 2018) or the recently published scientific color maps by Crameri (2020).

Secondly, a simple guidance on the necessity for color should be considered. The main uses of color are within these four categories:

A. distinguish categories (i.e., each color is a category),

B. visualize sequential data values (i.e., each color is a numeric value),

C. visualize diverging data values (i.e., each color is a numeric value and data have a direction and a meaningful midpoint or centric value), and

D. highlight some categories as special case of category A; i.e., a few categories have a color and the remaining ones have a grey gradient.

Moreover, there are also special cases of categories A–D such as multi-sequential color maps (e.g., bathymetry and topography with a centric value but not diverging) and circular/cyclic color maps (e.g., river orientation, with repeating colors, e.g., 0 and 360°) as outlined in Crameri et al. (2020).

We recommend as a first step to assign used data to one of the four categories A to D. If this is not possible, color might not be the best way to convey the data. Our survey revealed that especially the cases B and C were often mixed up or used color in the wrong way (Fig. 7). Examples of how to find appropriate color maps based on data types can be found in the literature (Coalter, 2020; Light and Bartlein, 2004; Zeileis et al., 2019; Crameri et al., 2020). Empirical assessment of quantitative color maps has also shown that diverging color maps (e.g., blue–white–orange) are slower and more error-prone during data encoding compared to single-hue color maps (Liu and Heer, 2018). Here comparisons of data values across the white midpoint are critical. For example, there is no need to have diverging color maps to visualize

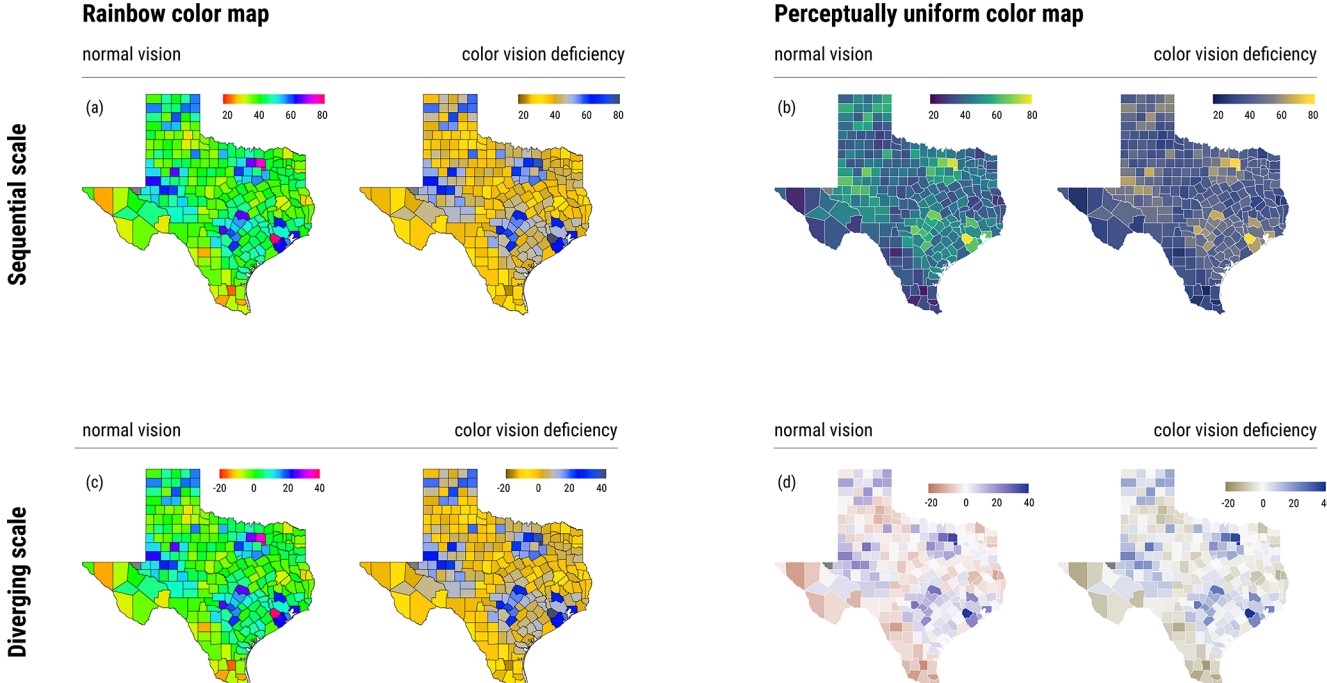

**Figure 7.** Comparison of mapping with rainbow (first column) and perceptually uniform color maps (third column) for an arbitrary variable across the counties of the US state of Texas. For each map, the vision for people with deuteranopia (red–green blindness) is given (second and fourth columns). With the rainbow color map, values around 60 and 20 stick out **(a, c)** and greenish or yellowish colors dominate a wide range of data variation. Compared to that, uniform and colorblind-friendly color maps **(b, d)** support a more exact perception of data variation and the extreme values. White strokes between the map elements **(b, d)** increase the data–ink ratio. Maps with emulation of color vision deficiency were created with the R package colorblindr.

correlation coefficient between 0 and 1, but they may be useful for ranges between −1 and 1. However, diverging color maps are meaningful and efficient as long they have a meaningful midpoint or centric value (often zero, mean value or 50 % value) and are used to highlight the direction of data or the data extremes, where the midpoint representation is damped, for example, with light grey color.

Another important point is to know typical color maps of specific types of visualization or in specific research areas (Gomis and Pidcock, 2018). For hydrology or environmental science, the use of specific diverging color scales such as red/blue for temperature or green/brown for aridity can also function as a first and easy decodable signal for the reader to give advice on what is actually shown in the presented figure or map (Fig. 7). Text elements or pointers will support colored data decoding for people with low vision. Such thematic and discipline-specific color choices aligned with peer discussions should be part of a thoughtful visualization development within each author team.

## 3.3 Trust color – tools to check for colorblind-friendly visualizations

"In perceptual-uniformity we trust!" (Crameri, 2017). There are mainly two possibilities to increase the trustworthiness of color choice in scientific visualizations. Firstly, by learning from surveys testing the trustworthiness of colors and different color maps. Examples are the Marie Skáodowska-Curie (Crameri et al., 2020) or the Which Blair project (Rogowitz and Kalvin, 2001), where different color palettes are mapped to well-known photos to highlight the effect of different color choice. These studies showed with experiments that color maps not based on a monotonically increasing luminance component produced no positive rating scores (Liu and Heer, 2018). Considering these findings, tools like ColorBrewer 2.0 (https://colorbrewer2.org, last access: 29 July 2021) give color advice on what kind of single- or multi-hue color maps can be used for sequential, diverging or qualitative data and let the user explicitly filter for colorblind safe and printer friendly color maps (Coalter, 2020; Harrower and Brewer, 2003). With the R package colorspace (Zeileis et al., 2019), a set of over 80 color palettes can be visualized and compared among each other. In the R package scico (Pedersen and Crameri, 2020) users will easily find palettes that embed common colors for visualization in their disciplines (see Sect. 3.2). The R package RColorBrewer (Neuwirth, 2012) provides color maps for visualizations and offers the possibility to use only colorblind-friendly color maps.

Secondly, when a visualization is ready for publication authors should aim for testing the specific figure or map to

**Table 1.** Checklist to improve color encodings in data visualizations. Tasks (1–4) are assigned to different roles during the publication process: author, co-author, reviewer, editor, journal and audience. Single actions are classified in recommended (R) and/or advanced (A). All URLs in this table were last accessed on 29 July 2021 TS2.

| Role | Level | Action | Reference and further reading |
|---|---|---|---|
| **1. Avoid color** | | | |
| Author | R | Revise graph type to remove or reduce color encodings; | Wong (2011a) |
| | R | Add second data encoding (e.g., point shapes) to resolve color ambiguity; | Kelleher and Wagener (2011) |
| | A | Use subgraphs or facets to focus attention and to reduce dimensionality; | |
| | A | Improve data–ink ratio by removing clutter (e.g., grid lines, background colors). | Tufte (1983); Gomis and Pidcock (2018) |
| **2. Improve color** | | | |
| Author | R | Use the right color map according to data type. Be sure that sequential and diverging color map is not confused! | Crameri et al. (2020) |
| | R | Replace rainbow color map using https://colorbrewer2.org or software; | Wong (2011c, b) R and Python tools: Sect. 3.2 |
| | A | If customized color map is needed, consider professional tools and CVD accessibility; | http://vrl.cs.brown.edu/color https://coolors.co |
| | R/A | Use discipline- and variable-specific color encoding. Clarify and simplify graph with annotations, direct labeling, binned color map and use color to highlight elements. | Gomis and Pidcock (2018) |
| **3. Trust color** | | | |
| Author Co-author | R/A | Inform yourself about trustworthy color palettes, e.g., Marie Skáodowska-Curie comparison and Which Blair project; | Crameri et al. (2020); Rogowitz and Kalvin (2001); |
| | R | Check your own work: run a CVD emulation to ensure accessibility of color; Check color contrast compliance or greyscale emulation. | see Sect. 3.2 for a list of tools https://contrastchecker.com |
| **4. Communicate color** | | | |
| Author | A | Add color statement: visualizations have been tested to be accessible for people with CVD (in papers, in presentations); | |
| | A | Consider adding a greyscale version of a visualization to the supplement; | |
| Reviewer Editor | R | Advise others on accessibility and potential deficiency of color maps; Consider color quality as review and decision criteria; | Crameri et al. (2020) |
| Journal | R | Provide clear color guidance including best practices on journal home page (i.e., in obligations for referees and editors, in the review criteria list); | |
| Audience | R | Raise awareness of color issues with your colleagues and other scientists; Give (friendly) criticism of rainbow color maps and propose alternatives. | Marie Skáodowska-Curie poster, Crameri et al. (2020) |

see how trustworthy the used color map actually is. With the R package colorblindr (McWhite and Wilke, 2021), various types of color blindness are simulated for production-ready R visualizations (Fig. 7). Open-source software applications for smartphones and computers offer a livestream of color-blindness emulation via camera or screen capture (e.g., https://github.com/michelf/sim-daltonism/, last access: 29 July 2021). Such emulations are also available on the internet without using a specific software, e.g., https://www.color-blindness.com (last access: 29 July 2021) or http://hclwizard.org (last access: 29 July 2021). Figure files can be uploaded to compare different visions such as normal, deuteranopia, tritanopia or monochromacy. To improve visualizations for people with other vision impairments, online tools like https://contrastchecker.com (last access: 29 July 2021) help to test color contrast compliance or to perform

greyscale emulation. As a last step the R package paletteer (Hvitfeldt, 2021) offers the possibility to produce some descriptive statistics on a color map or publication-ready figure and to run routines that optimize the color map, e.g., avoid colors that are appearing too similar. Finally, colorblind-friendly color maps are not the answer to everything. Authors should still check for sufficient design of visualization elements as point sizes are often too small, line widths are often too thin or overplotting impedes a full view of the data.

## 3.4   Communicate (rainbow) color – what should scientists and publishers do?

Literature review and our paper survey of color maps in scientific publications suggest that there is a considerable discrepancy between what science knows about the rainbow color map and what scientists do about it. The presented paper survey shows this discrepancy. Around 46 % of published *HESS* papers in 2020 have color issues (Fig. 2), but at the same time only less than 4 % of all reviewer comments see the choice of color as an issue before publication. From a knowledge perspective, the rainbow color map distorts a correct representation of the data variation. Thus, a reliable scientific communication is not possible with the rainbow color map. This knowledge is not reflected in the submission process of many journals, although the journals have been requested to raise author's awareness about CVD accessibility issues (Albrecht, 2010). If the extensive use of rainbow color maps in science continues, then also journalists and (social) media will most likely continue to circulate those rainbow figures and maps (Moreland, 2016). Today rainbow thumbnails appear in graphical abstracts, as thumbnails on journal websites or as screenshots in paper announcements on Twitter. This suggests for a broader audience that the rainbow color map is state of the art and scientifically correct as the visualizations have been produced by scientists with a high reputation in the public opinion. For young(er) scientists, the high reputation of scientific journals justifies rainbow color maps as appropriate for their own scientific work.

To leave this vicious circle, a major effort in science communication is needed. Authors can actively state in their papers that no rainbow color maps are used and visualizations have been subjected to a color accessibility check. Publishers and editorial teams should review graphical abstracts and summary thumbnails for rainbow color maps as authors tend to use rainbow-colored figures from the paper to attract readers' attention on websites with paper previews. Journal's author guidelines should specifically advise against the use of rainbow and red–green scales. Editors and reviewers should ask for revisions of rainbow figures and should be more relentless here. At scientific conferences, short courses for improved data analysis or environmental visualizations should raise awareness for the rainbow color map topic, especially for young(er) scientists. All of this is not about blaming the authors of rainbow visualizations but to clearly criticize those figures and mappings in a fair and constructive way, proposing methods to improve or avoid the (rainbow) coloring. Although the rainbow color map has more or less a tradition in various hydrological subdisciplines (e.g., in visualizations of water velocities, heat or solute transport as well as cartographic maps in general), we especially encourage the networks of young scientists to take responsibility for visualizations with valid color maps and a clear undistorted message. Communication of rainbow flaws should take place in all areas of science: during lectures, with colleagues, in network meetings, as feedback for presentations, as a conference attendee or paper reviewer but also as a journal editor, senior scientist or professor. The anti-rainbow Marie Skáodowska-Curie poster from Crameri et al. (2020) is freely available and could be a communication starter at the wall near the coffee machine of your institute.

## 4   Conclusions

The rainbow color map attracts attention but distorts and misleads scientific visualizations. Major rainbow pitfalls are the non-linear data encoding, steps and disorder in luminance, and minor perceptual accessibility for people with CVD or other vision impairments (Figs. 1 and 7). Here we investigated the use of rainbow color maps in around 1000 papers in different environmental journals and found that the misleading rainbow color map or red–green color issues are present in around 44 % of all papers (Fig. 2). We found no journal-specific differences in the use of the rainbow color maps (Fig. 3). Compared to the knowledge about the flaws of the rainbow color map, this share is alarmingly high. Moreover, our hypothesis that rainbow color maps are on the decline could not be confirmed. Color issues in papers remained constant or even increased between 2005 and 2020 (Fig. 2). Multi-author papers are not less prone to (rainbow) color issues, even though more people could weigh in against inaccessible visualizations (Fig. 4). Analysis of reviewer comments highlights that the awareness of those issues is alarmingly low during the review process.

Our survey indicates that past campaigns to banish the use of rainbow color maps were not sufficient. We strongly recommend that this issue should be raised across the hydrologic community. It will take students, researchers, lectures, professors, editors, reviewers and publishers to banish the rainbow color map, as well as simultaneous red and green usage, to make visualizations accessible for all readers and to insist on correct data representation. As a guide we presented manifold visual techniques on how to avoid, improve, trust and communicate color in data visualizations (Sect. 3.1–3.4, Table 1). Such guidance is given with a focus on important graph types in hydrology to attenuate the role and risks of color use in data encoding. Visualizations could not only be improved for people with CVD but should be drafted with more care in terms of less exclusive data en-

coding by color (Fig. 6). Such efforts could also bring advantages for a much larger group of people with low vision or vision impairments if more focus is given to visualizations with less clutter, higher contrast and supportive graphical elements like annotations.

## Appendix A

**Table A1.** Survey of papers in the *Hydrology and Earth System Sciences* journal. Classification is done based on expert judgment. Missing percent up to 100 is due to rounding. In 2020 only papers that were published before 1 November are considered.

| | Year | 2005 | 2010 | 2015 | 2020 | Total |
|---|---|---|---|---|---|---|
| Class | Description      Number of papers | 54 | 191 | 289 | 263 | 797 |
| A | Pure black–white | 44 % | 18 % | 4 % | < 1 % | 9 % |
| B | Chromatic without color issues | 35 % | 41 % | 48 % | 53 % | 47 % |
| C | Red–green color issues | 9 % | 16 % | 22 % | 23 % | 20 % |
| D | Rainbow color map issues | 11 % | 25 % | 26 % | 24 % | 24 % |

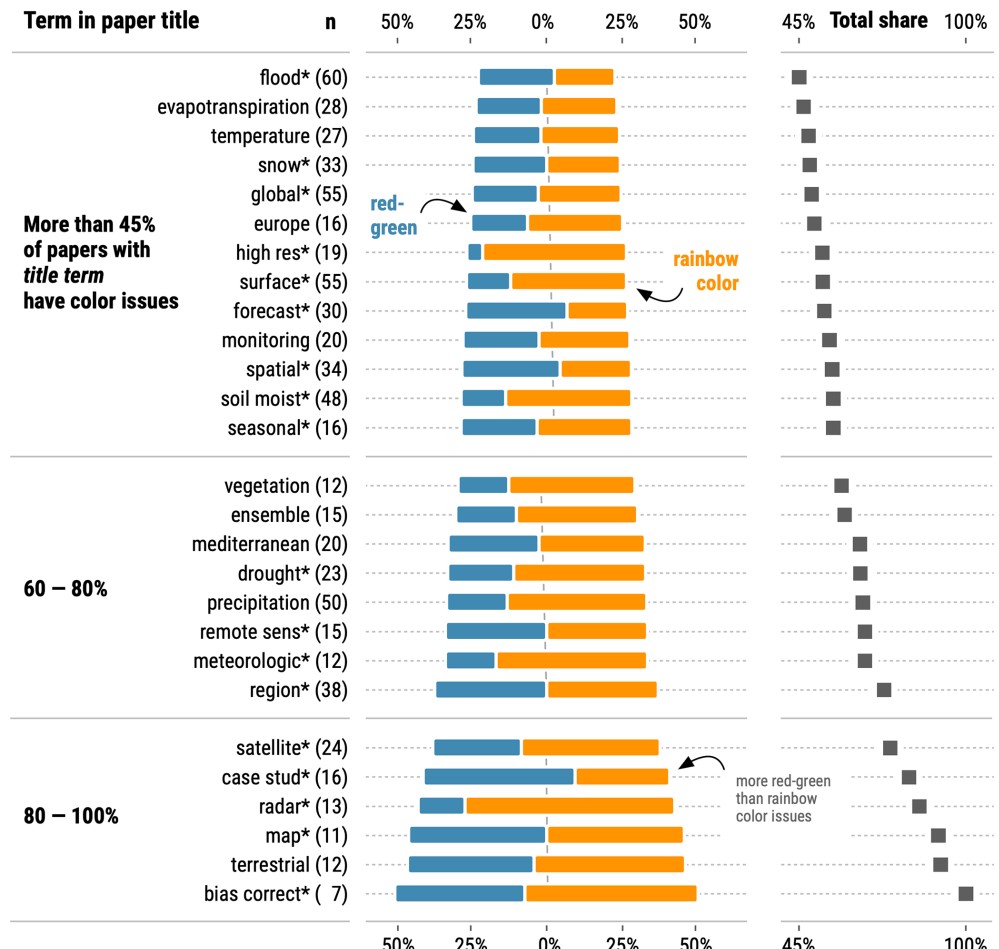

**Figure A1.** Text mining analysis finding 27 groups of papers sharing the same title term and having an above-average share of color issues. A paper title is only considered for analysis if the group of papers with the specific term in the title has more than 44 % of papers with color issues (above the average of 797 papers). Asterisk (*) acts as a wildcard in the regular expressions during text mining, e.g., with *map\** also the terms *maps* and *mapping* are searched. Listed terms have a minimum of five letters (exception *map\** and *snow\**) and occur in at least 10 individual papers (exception *bias correct\** with 7 papers). A specific paper could be part of multiple paper groups. More generic words like effect, approach, change, based, water or model and country names like China or France were excluded by expert judgment. Analysis is based on all 797 surveyed papers from the journal *Hydrology and Earth System Sciences* (2005–2020). The median author number across all groups of papers with the same title term ranges between 3.0 (*map\** and *seasonal\**) and 5.0 (evapotranspiration, Mediterranean, *region\**, and *soil moist\**).

*Data availability.* Data of the paper survey and sample code snippets are available online at https://github.com/modche/rainbow_hydrology (Stoelzle, 2021a) and as Kaggle Notebooks for exploration at https://www.kaggle.com/modche/rainbow-papersurvey-hydrology (Stoelzle, 2021b).

*Author contributions.* MS developed the research idea and designed the paper survey. MS and LS carried out the paper survey and analyzed the results. MS wrote the article and designed the figures with contributions from LS.

*Competing interests.* The authors declare that they have no conflict of interest.

*Acknowledgements.* Robin Schwemmle helped to evaluate the color issue classification, and Angelika Kuebert suggested R packages for colorblind-friendly color mapping. Thanks are due to Robin Schwemmle and Markus Weiler, whose comments greatly helped to improve the visualizations and the paper. Some visualizations are inspired by the book Fundamentals of Data Visualization by Claus O. Wilke (online available on https://clauswilke.com/dataviz/, last access: 30 July 2021). Authors also thank Fabio Crameri, Thorsten Wagener and an anonymous reviewer for their comments, which have helped to improve the paper. A community comment from J. C. Refsgaard is also appreciated.

*Financial support.* This open-access publication was funded by the University of Freiburg.

*Review statement.* This paper was edited by Jim Freer and reviewed by Fabio Crameri, Thorsten Wagener and one anonymous referee.

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

**Remarks from the typesetter**

TS1    Content changes like new values require the approval of the editor. Please give an explanation of why this needs to be changed. We will forward this request to the handling editor of this paper. Many thanks.

TS2    Please note addition

TS3    Please provide a date when you last visited the website (dd/mm/yyyy).

TS4    2019 was left as the new reference provided was without a year. Please confirm.