# Peer review of "Rainbow color map distorts and misleads research in hydrology – guidance for better visualizations and science communication"

_Hydrology and Earth System Sciences, 2021_

## Author Response (AR1)

**Letter of response**

*Stoelzle, M. and Stein, L.: Rainbow colors distort and mislead research in hydrology – guidance for better visualizations and science communication, Hydrol. Earth Syst. Sci. Discuss. [preprint], https://doi.org/10.5194/hess-2021-118, in review, 2021.*

Dear Editor,

We would like to thank the three reviewers, the community comment and the editor for the feedback on our manuscript. Please find our detailed answers to the reviewers' comments and the changes we have made below (responses in bold font). A track change version of the manuscript is also uploaded. We also decided to modify the paper title a little bit for clarity: "*Rainbow color map* distorts and misleads research in hydrology – guidance for better visualizations and science communication".

Best regards,

Michael Stölzle and Lina Stein

**Reviewer #1**

1. Figure 6: The figure could be made more intuitive by clearly labelling the top row of panels too (which are currently not labelled). It should be clear (either by additional on-figure text or via the caption and added panel labels) that the ‚original' panel is the not suited version to improve, and the three panels to its right-hand side are options to make it suitable. So, I suggest to add (a), (b), (c), … labels to all panels or clarify the on-figure explanation, e.g., by „story-telling graph titles". **Thanks for this comment. We revised the Fig. 6. Detailed answer can be found in comment #53 from Reviewer #2.**

2. lines 257-259: These points were also made in Crameri et al., 2020 (already in reference list of the current manuscript), to which the reader could be pointed to here as well. **Reference is added.**

3. lines 310-311: The reference for the actual Scientific colour maps is: Crameri, F. (2018). Scientific colour maps. Zenodo. http://doi.org/10.5281/zenodo.1243862 **Reference is corrected now.**

4. lines 312-318: The authors might consider clarifying that there are two other potential data types/options (which however might be less common): multi-sequential (e.g., bathymetry + topography with a centric value but not diverging) and circular/cyclic (e.g., river orientation; with repeating colours for e.g., 0 and 360°) as outlined in Crameri et al., 2020. **Thanks, we completed the list.**

5. line 320-321: This is a valid point to make. **Thanks**.

6. line 322: Crameri et al. (2020) provides a handy flow chart to select a colour map based on the data to be visualised, which could be referred to here as well. **Yes, we added the reference.**

7. lines 343 and 391: I suggest to change: „Marie Curie" to „Marie Skáodowska-Curie" **Changed**.

8. line 341: The reference for „Shephard et al., 2017" should be:
Crameri, F. (2017), The Rainbow Colour Map (repeatedly) considered harmful, edited by G.E. Shephard, EGU-Geodynamics Blog, https://blogs.egu.eu/divisions/gd/2017/08/23/the-rainbow-colour-map/, last access: 13 January 2021 or something like that, as it was written by myself and edited by Grace Shephard without contribution from others.
**Corrected now, thanks for clarification.**

9. line 374: A good reference to back up this statement would be:
Moreland, K. Why we use bad color maps and what you can do about it.
Electron. Imaging 2016, 1–6 (2016).
who concludes that the widespread use of the rainbow is the main reason scientists (and others) propagate it further.
**Yes, indeed, this fits well and is included now.**

10. line 405: Not to leave out the other important aspect investigated here, the authors may rewrite to: „…to banish the rainbow color map, and simultaneous red and green usage, …" or something along these lines.
**Good point, we added this to sentence.**

11. lines 127-128: That sentence sounds unclear to me; consider clarifying. E.g., the term „vision deficiency scale" sounds somewhat arbitrary.
**Scale is misleading here, we clarified the sentence now.**

12. line 132: If it is actually the case, consider clarifying that it means „all papers published in HESS".
**Yes, this is right, now clarified.**

13. line 212: Consider informing the (potentially non-hydrologist) reader that the following suggestions are specific to/from the field of hydrology, as some of the used terms (e.g., ‚response surfaces') are likely not familiar to readers from outside the discipline, and a potential source of confusion.
**We aimed to give also illustrative examples specifically for hydrologists how to learn from black and white visualizations, hence some discipline-specific terms are given in this section. However, we simplify some phrases here.**

14. line 234: „point" instead of „points"
**Corrected.**

15. line 318: Point D needs to be clarified grammatically.
**Corrected.**

16. lines 369-371: I do miss some critical commas throughout the manuscript. Here, for example, after „perspective" and „With that".
**Corrected**.

17. line 398: „alarming" to „alarmingly"?
**Corrected**.

**Reviewer #2**

18. The paper focusses mainly on color vision deficiency (CVD) and people with low/reduced vision, however one might also argue that good visualisation and labelling is equally important for people with other cognitive differences, such as (I'm guessing) dyslexia. Has there been research on this? If so, this aspect might be worth including in your literature review.
**The research in regard to cognitive difference and perception of data visualization is relatively limited (Lee et al, 2020;) and mostly focused on general data visualization and not scientific data visualization. While we recommend scientists take advice on making**

**their visualizations more accessible, the aim of our paper is the choice of color and less the choice of chart type. However, we have added a reference for further reading in Sect. 3.1: "Such efforts could be also beneficial for people with other limited (cognitive) capabilities (i.e., low visualization literacy or blind people) as there are possibilities of speech- or touch-based interaction with non-static visualizations (Lee et al., 2020)."**

19. The discussion of color palette type (negative-to-positive, strictly negative, or strictly positive) comes a little late in the manuscript (Figure 7c-d). It might be worth describing the type of color gradient that is most suited for negative-to-zero, negative-to-positive, and zero-to-positive scales sooner; e.g. a red-white-blue palette, which is currently missing from Figure 1.
**We see this point, too, but decided to recommend firstly techniques to avoid (or reduce) the use of color and THEN techniques to improve color. Therefore, the introduction of color maps is placed in Sect. 3.2. A changed order would challenge the meaningfulness of the "Avoid color"-technique. However, we added a diverging color map to Fig. 1 to complete the list of available color maps there and we added more explanation about the diverging color map in Sect 3.2**

20. It might be helpful to provide the readers with a "checklist" of items to verify when creating a readable scientific figure (e.g. "the data-ink ratio"; "a white mid-point at zero for negative-to-positive palettes").
**Yes, we added this checklist as a guidance table to the paper.**

21. Some repetition could be avoided, e.g. section 3.4 also contains some repetitions about CVD etc; perhaps it could be condensed a little.
**Yes, there was a repletion and we shorten the Sect 3.4 accordingly.**

22. It was useful to read about the colorblind options in R packages. Are there similar options for Python users?
**Yes, there are, we added a reference for the** `seaborn` **library in Python (see Section 3.2).**

23. I wondered if the paragraph about preprints (l.89-95) was really useful. It seemed to me this was a small sample compared with the analyses in subsequent paragraphs, so the utility wasn't entirely clear.
**We used the preprint analysis later on for the comparison between rainbow color maps in preprints and in published (peer-reviewed) papers. With that, we investigated the effect of the review process as we hypothesised that the review process should decrease the number of publications with color issues. However, we shorten the preprint paragraph, as it is a smaller analysis compared to the main survey.**

24. Lines 212-218 and elsewhere mention various types of visualisations (e.g., heatmaps at l.252), but it might be helpful to see examples (especially examples of good hydro-climatological visualisations).
**Indeed, that would be nice but also will increase the length of the paper. In the mentioned paragraph nine references are given (mostly open access papers) where the reader can find examples of good hydro-climatological visualizations. However, in the case of heatmaps we added a more detailed reference (Supplementary Figure 3 in Crameri et al., 2020).**

25. Finally, the title focusses on the hydrologic community but there were large parts of the text that were not specifically hydrological. Perhaps this could be strengthened a little. For example, Figure 1 could provide examples of hydroclimatic variables (i.e. highlighting which types of palettes are particularly suitable for specific variables).
**The line chart and mapping examples are directly designed to match hydrological science. However, we will add a reference to the IPCC Visual Style Guide for Authors (Gomis & Pidcock, 2018) where, for example, typical color maps for temperature or precipitation data are presented.**

26. l.14 raise awareness how -> raise awareness of how
   **Corrected.**
27. l.14 the rainbow color maps still is -> the rainbow color map still is
   **Corrected.**
28. l.23 we sketch a way to improve the communication of rainbow flaws -> we outline an approach to ?? (unclear what is meant by 'improve the communication of rainbow flaws')
   **We shorten the abstract here and revised the sentence to make clear that different status groups in science could act on avoiding the rainbow color map and also how they can name the flaws of the rainbow color map.**
29. l.31 10 millions of unique -> 10 million unique
   **Changed to "ten million unique colors"**
30. l.35 In terms of correct encoding, (comma needed for meaning)
   **Corrected.**
31. l.36 "we are stronger in encoding..": meaning could be clarified
   **Yes, we revised the sentence.**
32. l.42 "uses" -> "used"
   **Changed.**
33. l.47 the word "shares" is used instead of "percentages" (here and elsewhere); perhaps consider replacing for clarity
   **Thanks for the suggestion, but we think the meaning of 'share' is clear for the reader.**
34. l.61 the term "perceptual uniform" needs to be explained. I would recommend replacing "perceptual uniform" with "perceptually uniform" throughout the paper. It is explained at line 309, but this comes too late.
   **Thanks for this important comment. We added a definition now in the Introduction ("In perceptually uniform color maps, the delta change in color is equal to delta change in data") and changed the term consistently to "perceptually uniform".**
35. l.87 notable -> notably
   **Corrected.**
36. l.114 "a graph with two lines encoding continuous variables over time without any annotations… is classified as rainbow-related": worth providing examples alongside A-D for clarity?
   **Yes, we referenced to Fig. 6 to explain the line graph example.**
37. l.127 "a vision deficiency scale" – terminology could be clearer.
   **Yes, see also comment #11 from Reviewer #1 – now corrected.**
38. l.139 "two cross checks… led (not lead) to minor deviations": if this information is included, then it might be worth specifying what "minor deviations" means and how many people are in the cross-checking and original reviewer teams.
   **Minor deviations (rainbow paper classification mismatch during two cross-checks: 14% and 8%) are explained in the following sentence in the manuscript. We were 3 persons in the survey/reviewer team.**
39. l.149 It might be worth justifying the choice of journals – why were Sci Rep and NComms selected?
   **We justified the selection by the reputation of the journals and the fact that in both journals papers from different disciplines are published. We added this information to the paper (Sect 2.2).**
40. l.168 "a current redistribution of disappearing black and white papers into papers with and without color issues". I think this means something like 'coincidence between the decline of black and white papers and the emergence of papers with color issues'
   **Thanks for this comment, we replaced the unclear sentence with this suggestion.**
41. l.186 less -> fewer
   **Omg, yes.**
42. l.190 73-92% of how many? A little unclear why two numbers here.
   **This is visible in Fig. A1. We clarified the sentence and added a reference to the figure.**
43. l.193 four "suggestions" perhaps
   **Corrected.**

44. l.233 ScientistS
   **Corrected.**
45. l.237 "a pointedly use of" is unclear
   **Yes, Changed to "targeted".**
46. l.246 "luminance" is unclear (also used elsewhere in manuscript). Does it mean transparency? Shading?
   **Luminance is the photometric measure of how much light per area is apparent. It is not the same as transparency or opacity. Brightness is the subjective perception (often scaled in percentages).**
47. l.384 rise awareness -> raise
   **Corrected.**
48. l.289 parts of science -> areas of science
   **Ok, corrected.**
49. Figure 1. "The same delta changes in values": this could be rephrased for clarity; it is not entirely clear what the "+1" on the figure or in the caption refer to. Also, is "perceptual uniform" "perceptually uniform"? By this point in the manuscript (line 70), I think if would be helpful to distinguish the colors used for scales that range from negative to positive (e.g. "red white blue") and those that are "strictly positive" or "strictly negative".
   **Figure is revised accordingly.**
50. Figure 2. I wonder if it might make more sense to show the % of red-green or rainbow color maps as a fraction of the total number of papers (instead of just the papers with color issues).
   **We think the searched information is already there: Percentages of red-green-papers and rainbow papers are given for the years 2005, 2010, 2015 and 2020.**
51. Figure 4 seems clear to me.
   **Ok.**
52. Figure 5 is a little unclear. I wonder if examples (of alternatives, properties, tools etc.) could be provided for clarity?
   **Thanks for this comment. We extended the figure caption to provide more information for the reader.**
53. Figure 6.Dark background – is this supposed to be easier or harder to read? Panel c. is it brightness or shades? Panel i. beyond the graph title, good labelling can also be helpful. Historically, many journals have discouraged the use of labels on figures; but for some people, clear panel/facet labelling can help greatly. Perhaps this is worth a mention. Also worth making sure that all panels are referred to in the main text.
   **Dark background can be a valuable technique to increase contrast of charts during presentations (information is given in the caption of Fig. 6). Yes, in panel c. "line brightness" is right. Panel i.: Yes, good labelling is also helpful – as we mentioned in panel e. and f. of the same Figure. All panels are referred to in the main text now.**
   **We also added numbers (1-4) and some arrows to guide the reader through the figure and added a "2. CVD / greyscale version" to make even more clear why the Techniques in 3. are needed. We revised the caption of Fig. 6.**
54. Figure 7. OrangeRed and Batlow are almost too small to read; would recommend deletion.
   **Yes, deletion of the subpanels led to a clearer figure now.**
55. Do you mean "white strokes decrease the data-ink-ratio" (rather than increase)?
   **No, with white stroke lines in the map (=no ink) the data-to-ink ratio is increased (as ink is decreased). With black stroke lines (=ink) this ratio would be lower.**
56. Is the correct technical term "color map", "color scale", "color palette", or "color gradient"?
   **Very good comment. This is not consistent throughout the literature. "Color gradient" might be really misleading but "palette" is also often used (more in the manuals of different software packages.) Our impression was that color map is the best term as it also reflects the important task of mapping data to color. The term "color map" is thus consistently used in the paper.**

**Reviewer #3**

57. Maybe I missed this in the paper, but how many software packages (Matlab etc.) offer a rainbow colour scheme as the default? Do the authors simply use the default and not think about it? This would be my personal hypothesis based on my own past mistakes. If many software packages offer this as default scheme, then is there just a straight mapping of default schemes and schemes used? If so, then would the best strategy to approach the software producers to change their schemes (rather than focus on the users)? How much does the use of rainbow colour schemes correlate with the default colour scheme in the software used (do the authors have the data to calculate this)?

**This is a very interesting point. We do not have a data set on the correlation between rainbow color map occurrence and the used software to produce the corresponding visualizations. But from our paper survey it is obvious that authors often use the standard color scheme, e.g., in R statistics to generate graphs such as line or point plots. As we mentioned in L41-43 the color preset of R version 3.x uses "black, red, green and dark blue" for the first four data sets, and color confusion is hence a logical consequence. MATLAB changed its default color system in 2014. Instead of blue, green, red and turquoise for the first four data points the colours are now blue, orange, yellow, purple. And instead of "jet", the MATLAB rainbow equivalent, "parula" is used. However, "parula" is not considered as 100% perceptual uniform either as the luminance gradient is steeper at the edges and flatter in the middle of the color map. The parula color map could be found in a lot of papers from our survey. Same presets might exist in other software or visualization products/tools. However, advocating the companies to change a non-colorblind safe preset might force authors to implement their former choice as they are used to it. We assume that often the reason for the fame of visualization with the rainbow color maps is – as we wrote (L38-40) – that these figures look really colorful, appealing and attract attention. We guess that the authors also think that rainbow color boost their visualizations in a way to be more impressive or outstanding. However, a lot of software products and programming languages offer the possibility to load your own color maps. In other words, we think you cannot forbid the scientist to use a specific color map, but we might have possibilities (or at least make some efforts) to forbid the authors to publish papers with rainbow visualizations (and also red-green-figures) as the rainbow color map is considered to be scientifically incorrect.**
**Finally, we added a recommendation to the guidance table to ensure that the used color map is colorblind-friendly and perceptually uniform (see "Trust color" in the table)**

58. The authors discuss in section 3.3. that tools like colorbrewer2.0 and others can be used to avoid issues for colour blind people. Tools like these offer a much wider help to avoid a wide range of colour issues discussed in this paper. Do the authors not think that a general use of such tools would avoid most errors they discuss? Basic use of such tools for all colour choices would solve most of the problem, why not suggest this as a standard? Would this be easier than a list of things that the scientist has to check separately?

**Thanks for this important comment. Yes, it seems to be reasonable to add a precise statement on potential standard tools (like colorbrewer2.0). We have revised the Section "Improve color" accordingly and have also revised the Section "trust color" arguing that a cross-check (e.g., with a CVD emulator) helps the authors to be more confident about the use of color in their visualizations. If, above that, a own color map must be created we recommend tools for that, too.**

59. Point two leads me to my third point. The authors state at the end of their paper that "As a guide we presented manifold visual techniques…". This is great for those highly motivated to do the right thing in terms of publishing visualizations, but there is a risk that this will be too much for many scientists. Is there a simpler step-wise guide the authors could propose? My personal strategy is to require all my students to use colorbrewer 2.0 to ensure that major errors are avoided, but maybe the authors could summarize their suggestions into a few key points?

**We added a step-wise guidance table/checklist (Table 1) for the reader.**

**Literature**

310    Gomis, M. I. and Pidcock, R.: IPCC Visual Style Guide for Authors, IPCC WGI Technical Support Unit, 28, 2018.

Lee, B., Choe, E. K., Isenberg, P., Marriott, K., & Stasko, J. (2020). Reaching broader audiences with data visualization. IEEE computer graphics and applications, 40(2), 82-90.

---

## Author Response (AR2)

7 July 2021

**Authors response to**
**Editor Decision: Publish subject to technical corrections (30 Jun 2021) by Jim Freer.**

Dear Editor,

Thank you for the two suggestions and the chance to publish our work in HESS.

Firstly, we have integrated your smaller suggestion (i.e., greyscale version of graph to supplement) into the checklist table 1.

Secondly, and more importantly, we have drafted a new version of this checklist (Table 1). We have discussed several possibilities and now decided to have a rather short version of the table that hopefully can have a consolidated design in the typeset version of the manuscript, As you can see, we have now four columns in the table (roles, levels, task and references) and implemented the four sections (3.1-3.4) as sub-headings. We think we should now wait for the typesetting and then see how this table is best integrated into the end of section 3 (i.e., column-width or full-width).

Furthermore, there is a new publication from Christa Kelleher that should be mentioned in our paper. Thus, we have added a sentence to the beginning of Section 3:

```
Kelleher and Braswell (2021) published a comprehensive
introduction how visualization challenges of environmental data
can be assessed in a broader sense.
```

Thanks again for your efforts.

Best regards,
Michael Stölzle (on behalf of co-author Lina Stein)

Kelleher, C. and Braswell, A.: Introductory overview: Recommendations for approaching scientific visualization with large environmental datasets, Environmental Modelling & Software, 143, 105113, https://doi.org/10.1016/j.envsoft.2021.105113, 2021.